# Treatment of Asymptomatic Bacteriuria after Implementation of an Inpatient Urine Culture Algorithm in the Electronic Medical Record

**DOI:** 10.3390/pharmacy9030138

**Published:** 2021-08-11

**Authors:** Dianne Osiemo, Danny K. Schroeder, Donald G. Klepser, Trevor C. Van Schooneveld, Andrew B. Watkins, Scott J. Bergman

**Affiliations:** 1College of Pharmacy, University of Nebraska Medical Center (UNMC), Omaha, NE 68198, USA; dianne.osiemo@unmc.edu (D.O.); dklepser@unmc.edu (D.G.K.); 2Department of Pharmaceutical and Nutrition Care, Nebraska Medicine, Omaha, NE 68198, USA; dschroeder@nebraskamed.com (D.K.S.); anwatkins@nebraskamed.com (A.B.W.); 3Division of Infectious Diseases, UNMC College of Medicine, Omaha, NE 68198, USA; tvanscho@unmc.edu

**Keywords:** asymptomatic bacteriuria, urine culture, antibiotics, treatment

## Abstract

Ordering urine cultures in patients without pyuria is associated with the inappropriate treatment of asymptomatic bacteriuria (ASB). In 2015, our institution implemented recommendations based on practice guidelines for the management of ASB and revised the urine culture ordering process to limit cultures in immunocompetent patients without pyuria. The purpose of this study was to determine how the treatment of ASB has changed over time since altering the urine culture ordering process to reduce unnecessary cultures at an academic medical center. A quasi-experimental study was conducted for inpatients with urine cultures from January to March of 2014, 2015, 2016 and 2020. The primary outcome was the antibiotic treatment of asymptomatic bacteriuria for over 24 h. The secondary outcomes were the total days of antibiotic therapy, type of antibiotic prescribed and overall urine culture rates at the hospital. A total of 200 inpatients with ASB were included, 50 at random from each year. In both 2014 and 2015, 70% of the patients with ASB received antibiotic treatment. Antibiotics were prescribed to 68% and 54% of patients with ASB in 2016 and 2020, respectively. The average duration of therapy decreased from 5.12 days in 2014 to 3.46 days in 2020. Although the urine cultures were reduced, there was no immediate impact in the prescribing rates for patients with ASB after implementing this institutional guidance and an altered urine culture ordering process. Over time, there was an observed improvement in prescribing and the total days of antibiotic therapy. This could be attributed to increased familiarity with the guidelines, culture ordering practices or improved documentation. Based on these findings, additional provider education is needed to reinforce the guideline recommendations on the management of ASB.

## 1. Introduction

The presence of bacteria in urine with no symptoms of a urinary tract infection, known as asymptomatic bacteriuria (ASB), is a common occurrence that is often inappropriately managed with antibiotics [1]. The practice guidelines from the Infectious Diseases Society of America (IDSA) recommend against the screening and treatment of patients without symptoms due to a lack of clinical benefits, except in the following populations: pregnant women, patients undergoing an invasive urological procedure or kidney transplant recipients within 1 month of a transplant [2]. Despite the clear evidence-based recommendations, studies have found that 48–58% of patients with inpatient ASB are prescribed antimicrobial treatment inappropriately [3,4,5,6]. The factors that are associated with inappropriate prescribing for ASB include patient characteristics such as older age; acute altered mental status; dementia and laboratory characteristics such as pyuria, abnormal urinalysis, Gram-negative bacteriuria and higher bacterial colony counts [7]. Positive urine cultures are also associated with the treatment of ASB [8,9]. Without pyuria, bacteria in the urine are not indicative of infections in immunocompetent patients, and yet, urine cultures are often obtained in patients suspected of having any infection as part of a differential diagnosis [9]. This is often done in the emergency department for patients with an altered mental status as providers search for a diagnosis in those that they have very little information about. The urinalysis and culture results alone are not enough to diagnosis a urinary tract infection (UTI) without symptoms, yet antibiotic therapy that is started in the ED is often continued in hospitalized patients. An update to the IDSA Guidelines for ASB published in 2019 indicates no causal relationship between bacteriuria and an altered mental status. In the absence of genitourinary symptoms or other systemic signs of infection, no treatment is warranted [2].

In 2015, our institution implemented a urinary tract infection culture algorithm in the electronic medical record (EMR) as part of an initiative to reduce the rates of catheter-associated UTI [10]. The urine culture ordering process for inpatients was revised such that urine cultures could only be ordered using a UTI evaluation panel that required the documentation of UTI symptoms (e.g., fever, dysuria, urinary frequency, urgency, hematuria, flank pain, nausea/vomiting or costovertebral angle tenderness) or if special criteria allowing urine cultures in the absence of pyuria or symptoms of UTI were present (e.g., neutropenia, kidney transplant, pregnant or impending urologic surgery). The urine culture rates declined significantly from 35.9 per 1000 patient days (PD) to 19.3/1000 PD the year after the implementation in 2015 [10]. With reductions in urine cultures, it is expected that a corresponding decrease in antibiotic prescribing for UTI should be observed [11,12]. The purpose of this study was to determine how the treatment rates of ASB, antibiotic choices and overall days of therapy have changed over time since altering the urine culture ordering process in the electronic medical record.

## 2. Materials and Methods

This study occurred at Nebraska Medicine, an 809-bed tertiary care academic medical center in Omaha, Nebraska. The study population included inpatients > 18 years of age with positive urine cultures (>10^5^ colony forming units/mL) and no documented symptoms of a UTI admitted during four three-month periods (January–March) in 2014, 2015, 2016 and 2020.

The new UTI algorithm was posted to the antimicrobial stewardship program website (www.unmc.edu/asp) in 2014, and the UTI evaluation panel was implemented in April 2015. Therefore, these two years were considered the preintervention group, while 2016 and 2020 were considered after. Once the intervention occurred, inpatient urine cultures could only be ordered in the EMR for patients with both symptoms of UTI and pyuria or where a urine culture would be appropriate even in the absence of symptoms or pyuria. Education was provided to hospitalists, internal medicine residents and emergency department clinicians on how the algorithm works, as well as reinforcement of the institutional guidelines that recommend against the treatment of ASB.

Patients were excluded where the evaluation for and treatment of asymptomatic bacteriuria may be appropriate, including pregnancy, those undergoing a urologic procedure or those within 1 month of a kidney transplant. Patients with an indwelling catheter were also excluded, as symptoms of UTI are more difficult to assess, and the treatment guidelines differ. The primary outcome was an antibiotic treatment of asymptomatic bacteriuria for > 24 h. The secondary outcomes were the total days of antibiotic therapy, type of antibiotic prescribed and overall urine culture rates at the hospital.

This study was deemed exempt by the institutional review board. Based on the prior literature, we anticipated the treatment rate for ASB to be 50% and predicted a decrease in the treatment to 30% postintervention. A total sample size of 186 patients was required to have 80% power to detect an absolute difference of 20% in the treatment of ASB before and after the intervention (https://clincalc.com/stats/samplesize.aspx). A total of 200 patients were included, with a sample of 50 patients having asymptomatic bacteriuria randomly selected each year, as shown in Figure 1.

Hospitalized patients with a urine culture from each time period were randomly selected and screened for inclusion criteria. The charts were reviewed, and the patients were excluded if they had an indwelling urinary catheter or any documented symptoms of a UTI (fever, dysuria, urinary frequency, urgency, hematuria, flank pain, nausea/vomiting or costovertebral angle tenderness) at the time of the urine culture. The absence of symptoms of a UTI in the setting of a positive urine culture indicated that a patient had ASB (Figure 1). To determine the potential impact of the updated IDSA guidelines, we also analyzed how many patients were treated for a UTI based on their altered mental status alone. The antibiotic orders were evaluated to determine the indication and duration of therapy.

The proportion of patients with ASB treated with antibiotics before and after the intervention was compared using Fisher’s exact test and a Student’s *t*-test. The analyses were performed using statistical software SAS OnDemand for Academics (SAS Institute, Cary, NC, USA).

## 3. Results

The rate of urine cultures in patients that decreased immediately with the implementation of the UTI evaluation algorithm from 35.9 to 19.3/1000 PD remained significantly lower than the baseline in 2020 at 15.3/1000 PD.

After screening 1139 results of orders for a urine culture in the four quarters evaluated, 648 charts were reviewed before including a total of 200 inpatients with ASB for the study to evaluate the treatment. The patients were predominantly female and over 65, which did not change over time. The characteristics of these patients are further summarized in Table 1. The proportion of patients with an altered mental status as the sole reason documented for the treatment of a UTI increased each year and was significantly different in 2020 compared to 2014 (*p* = 0.028).

The results of the evaluation are described in Table 2. There was a 24% relative reduction in patients receiving treatment for ASB in 2020 from the preintervention time period (*p* = 0.07). Comparing the baseline treatment rate of 70% for 2014 to 2015 to the average of 61% for the postintervention years 2016 and 2020, no statistical difference could be detected (*p* = 0.23).

The average duration of therapy decreased from 5.12 days in 2014 to 3.46 days in 2020 (*p* = 0.015). With education to clinicians from the antimicrobial stewardship program to avoid fluoroquinolone use, especially for the empiric treatment of cystitis, a declining trend over the years was noted, with more preference for beta lactams, as seen in Table 3. (The patients that were changed from one class to another during therapy were counted twice in this analysis.) Fluoroquinolone prescriptions decreased from 34 per 100 patients in 2014 to 2015 to 17 in 2016–2020 (*p* < 0.05). There were 25 ceftriaxone prescriptions per 100 patients in 2014 to 2015 vs. 42 in 2016–2020 (*p* = 0.05). The shift away from fluoroquinolones in favor of beta lactams was especially apparent in 2020 with three prescriptions compared to 28, respectively.

Even after the introduction of order limits requiring the documentation of UTI symptoms, patient characteristics and criteria supporting urine cultures in patients without pyuria, urine cultures were still performed in 16-18% of ASB patients who did not fit the criteria. It was discovered that this was most commonly from the emergency department calling the microbiology lab to request a urine culture rather than using the revised order set in the EMR for a urinalysis with a culture performed only by reflex with pyuria that would have prevented them from getting a culture. 

## 4. Discussion

ASB is commonly treated inappropriately with antibiotics that contribute to an increased antimicrobial resistance, medication adverse events and collateral damage to the gut microbiota, leading to *Clostridioides difficile* infections and increased healthcare costs [3,13]. We demonstrated that the treatment rate for ASB at our institution in 2014 was higher than expected at 70%, and it decreased to the average in the literature five years after an intervention to reduce unnecessary urine cultures. The higher initial treatment rate may have impacted the power of our study to show a statistical difference in the primary outcome. Rather than a 20% absolute and 40% relative difference, we saw a 16% absolute and 24% relative reduction. This is similar to the reduction that has been reported in other studies [3]. We took a sample several years after the intervention, expecting to see a steady decline in treatments similar to the urine culture rates. However, the treatment rate remained the same after providing institutional guidance on the management of ASB in 2014, indicating no immediate effect of this relatively passive intervention. In 2016, the year after revising the electronic urine culture ordering process, a lot fewer urine cultures were performed, but the treatment rate for ASB remained similar at 68%. Five years after the intervention, there more of a decrease in the proportion of patients treated for ASB. Although this effect was predicted, it was not as immediate as expected. This could be due to the fact that patients less likely to be treated for ASB previously were the ones that no longer had urine cultures obtained. An additional sample between 2016 and 2020 would have improved the ability of our study to detect a statistical trend. The total duration of therapy for ASB also decreased over time, which indicates a likely improvement in the prescribing habits. A reduction in antibiotic usage should translate to reduced adverse events and the associated costs. The improvement over time in the reduced treatment and duration of therapy for ASB could be attributed to an increased familiarity with the culture ordering process and guidelines on the management of ASB.

When the overall decrease in urine cultures is taken into account, this reduction in the treatment is clinically meaningful. In 2014, there were 153,217 patient days at our hospital and 5654 urine cultures performed. In 2020, the hospital had decreased the number of urine cultures performed to 2320 on 158,896 patient days. We had to screen 1139 urine cultures altogether to find 200 asymptomatic patients without any indication for treatment or indwelling urinary catheters. If we apply this rate of 17.5% to the urine cultures in each year, there would have been 993 asymptomatic patients with cultures in 2014 and 408 in 2020. Based on the treatment rates of 70% and 54%, this would be 695 and 220 cases treated unnecessarily for a rate of 4.5/1000 PD in 2014 and 1.4/1000 PD in 2020, a 69% reduction overall.

These results are consistent with a systematic review demonstrating that computerized decision support has been found to improve antimicrobial prescribing practices [14] In a previous meta-analysis, the implementation of educational and organizational interventions designed to eliminate the overtreatment of ASB resulted in a median absolute risk reduction of 33% [3]. Ordering unnecessary urine cultures has been associated with the treatment of ASB, especially in patients without pyuria, and this can lead to antimicrobial resistance [11,15,16].

An aspect of our evaluation that could have changed the results would be withdrawing patients with an altered mental status from the classification of asymptomatic, since this was not specified until the 2019 IDSA guidelines were published. The education on these adjusted criteria had not been widely disseminated at our institution by the time of this study. Counting these patients as symptomatic and removing them from our analysis would mean 33 out of 48 patients (68%) were treated inappropriately in 2014 and 20 out of 40 (50%) in 2020.

Several limitations were identified in this study. The retrospective nature of this review relied on the documentation in medical records to differentiate ASB from UTI. The documentation process could have varied over the years, thus resulting in the possibility of some true UTI cases being considered as ASB. The sampling process and manual chart review may have introduced a selection bias in obtaining the medical records to be included in the study. As previously discussed, the sample size selected was small, thus resulting in a reduced statistical power and consequently reducing the chance to detect actual changes in the treatment rates for ASB. 

This study identified some key areas that need to be addressed to reduce the inappropriate treatment of ASB. Additional provider education is required to reinforce the IDSA guidelines on the management of ASB. A target audience would be emergency department providers, since it was identified that they bypass the urine culture ordering algorithm designed to limit the treatment of patients without pyuria. There is a high likelihood that antibiotic therapy initiated in the emergency department will be continued when the patient is admitted. Inpatient providers can be further educated on the updates in the IDSA guidelines to avoid using an altered mental status as a symptom of UTI when an alternative explanation is possible. Overall, our study demonstrated the challenges in implementing quality improvements involving antimicrobial stewardship at most medical centers.

## Figures and Tables

**Figure 1 pharmacy-09-00138-f001:**
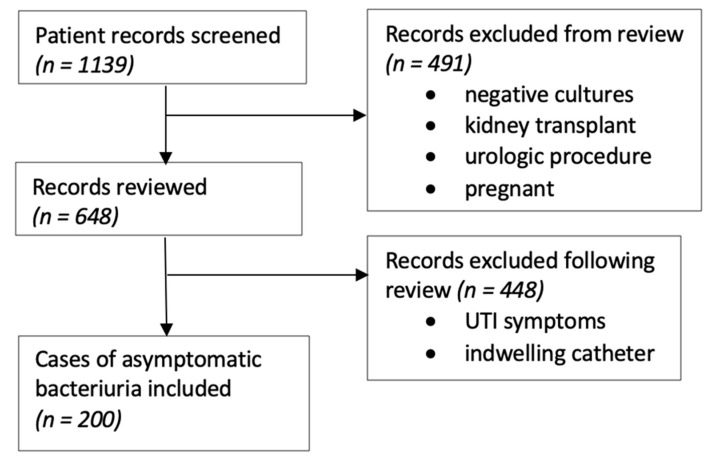
Flow diagram of the records reviewed to assess the urine culture algorithm intervention.

**Table 1 pharmacy-09-00138-t001:** Patient characteristics.

	2014 (*n* = 50)	2015 (*n* = 50)	2016 (*n* = 50)	2020 (*n* = 50)
Sex, no. (%)				
Female	43 (89%)	42 (84%)	43 (86%)	45 (90%)
Age > 65 y, no. (%)	40 (80%)	41 (82%)	43 (86%)	39 (78%)
WBC < 10, no. (%)	14 (28%)	8 (16%)	9 (18%)	8 (16%)
Altered mental status diagnosis, no. (%)	2 (4%)	6 (12%)	7 (14%)	10 (20%)

**Table 2 pharmacy-09-00138-t002:** Results.

	2014 (*n* = 50)	2015 (*n* = 50)	2016 (*n* = 50)	2020 (*n* = 50)
Treatment with antibiotics > 24 h, no. (%)	35 (70%)	35 (70%)	34 (68%)	27 (54%)
Mean duration of therapy (days)	5.12	5.83	4.96	3.46
Cumulative duration of therapy (days)	256	284	284	173

**Table 3 pharmacy-09-00138-t003:** The antibiotics prescribed to treat ASB.

Antibiotics Ordered, No.	2014	2015	2016	2020	Total
Fluoroquinolones	18	16	14	3	51
Beta Lactams	18	29	23	28	98
Trimethoprim/Sulfamethoxazole	8	11	8	9	36
Nitrofurantoin	2	9	1	7	19
Vancomycin	3	1	1	1	6
Fosfomycin	1	0	1	0	2

## Data Availability

The data presented in this study are available on request from the corresponding author. The data are not publicly available due to privacy restrictions.

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
