# Peer review of "Treatment of Asymptomatic Bacteriuria after Implementation of an Inpatient Urine Culture Algorithm in the Electronic Medical Record"

_pharmacy, 2021, doi:10.3390/pharmacy9030138_

Round 1
Reviewer 1 Report
Abstract:
Please explain all abbreviations (IDSA) on first mentio.
L36-38: Please include the following reference:
https://pubmed.ncbi.nlm.nih.gov/32596751/
L45-55: please elaborate more on this subject!
L66-69: please elaborate more on the purpose of the study
L71-73: Please include the following reference:
https://pubmed.ncbi.nlm.nih.gov/31324035/
L90-96: Please include the formula you based you calculation on.
Methods:
it is impreative that you provide data on your healthcare institution (technical details, patient turnover, scope of care etc.) as you reference these throughout the MS
Results: in general, the results section seems very devoid of results (or maybe they are inadequately described?). Please aim to provide a more detailed results section.
L174-182: please discuss the following paper, referring to the resistance rates detected at an Emergency Department (ED):
https://pubmed.ncbi.nlm.nih.gov/32961770/
Author Response
Abstract:
Please explain all abbreviations (IDSA) on first mentio.
I deleted the specific organization in the abstract to keep it brief, and then explained it in the introduction of the text.
L36-38: Please include the following reference: Added
https://pubmed.ncbi.nlm.nih.gov/32596751/
L45-55: please elaborate more on this subject! Done
L66-69: please elaborate more on the purpose of the study. Added more specifics
L71-73: Please include the following reference: Added to Discussion
https://pubmed.ncbi.nlm.nih.gov/31324035/
L90-96: Please include the formula you based you calculation on. Added
Methods:
it is impreative that you provide data on your healthcare institution (technical details, patient turnover, scope of care etc.) as you reference these throughout the MS.
Agreed. I included this in the methods.
Results: in general, the results section seems very devoid of results (or maybe they are inadequately described?). Please aim to provide a more detailed results section.
I revised this section to include more details
L174-182: please discuss the following paper, referring to the resistance rates detected at an Emergency Department (ED):
https://pubmed.ncbi.nlm.nih.gov/32961770/
Added
Reviewer 2 Report
This study is a retrospective review of ASB antibiotic prescribing outcomes pre and post implementation of electronic medical record UTI algorithm to limit urine culture ordering to UTI evaluation panel. This research is welcome addition as it goes beyond describing the impact of this on urine culture ordering.
Balanced discussion of study findings and potential implications.
Brief discussion how findings compare to published on the impact of computerised decision support tools could be considered. Eg Curtis CE, Al Bahar F, Marriott JF. The effectiveness of computerised decision support on antibiotic use in hospitals: A systematic review. PLoS One. 2017 Aug 24;12(8):e0183062. doi: 10.1371/journal.pone.0183062. PMID: 28837665; PMCID: PMC5570266.
Minor suggestions
- Suggest add “EMR” to title of submission – e.g. Treatment of ASB after electronic medical record implementation of an inpatient urine culture algorithm
- Cite reference for p 4 Lines 147-8“…it decreased to the average in literature five years after the intervention designed to reduce …” Suggest reference e.g. Myrto Eleni Flokas, Nikolaos Andreatos, Michail Alevizakos, Alireza Kalbasi, Pelin Onur, Eleftherios Mylonakis, Inappropriate Management of Asymptomatic Patients With Positive Urine Cultures: A Systematic Review and Meta-analysis, Open Forum Infectious Diseases, Volume 4, Issue 4, Fall 2017, ofx207, https://doi.org/10.1093/ofid/ofx207
- Typo error pg5, line 57 “facts” should be “fact”
Author Response
This study is a retrospective review of ASB antibiotic prescribing outcomes pre and post implementation of electronic medical record UTI algorithm to limit urine culture ordering to UTI evaluation panel. This research is welcome addition as it goes beyond describing the impact of this on urine culture ordering.
Balanced discussion of study findings and potential implications. Thank you
Brief discussion how findings compare to published on the impact of computerised decision support tools could be considered. Eg Curtis CE, Al Bahar F, Marriott JF. The effectiveness of computerised decision support on antibiotic use in hospitals: A systematic review. PLoS One. 2017 Aug 24;12(8):e0183062. doi: 10.1371/journal.pone.0183062. PMID: 28837665; PMCID: PMC5570266.
Minor suggestions
- Suggest add “EMR” to title of submission – e.g. Treatment of ASB after electronic medical record implementation of an inpatient urine culture algorithm – Good Point. Added
- Cite reference for p 4 Lines 147-8“…it decreased to the average in literature five years after the intervention designed to reduce …” Suggest reference e.g. Myrto Eleni Flokas, Nikolaos Andreatos, Michail Alevizakos, Alireza Kalbasi, Pelin Onur, Eleftherios Mylonakis, Inappropriate Management of Asymptomatic Patients With Positive Urine Cultures: A Systematic Review and Meta-analysis, Open Forum Infectious Diseases, Volume 4, Issue 4, Fall 2017, ofx207, https://doi.org/10.1093/ofid/ofx207
- Added
- Typo error pg5, line 57 “facts” should be “fact”
- Thank you for noticing, this has been corrected.
Round 2
Reviewer 1 Report
The authors have adequately addressed the major issues regarding the article. I warmly recommend the acceptance of the paper.